# ChebyNet: Boosting Neural Network Fitting and Efficiency through Chebyshev Polynomial Layer Connections

## Abstract

Traditional deep neural networks (DNNs) predominantly adhere to a similar design paradigm. Even with the incorporation of additive shortcuts, they lack explicit modeling of relationships between non-adjacent layers. Consequently, this paradigm constrains the fitting capabilities of existing DNNs. To address this issue, we propose ChebyNet, a novel network paradigm to build Chebyshev polynomial connections between general network layers. Specifically, we establish recursive relationship among adjacent layers and polynomial relationship between non-adjacent layers to construct ChebyNet, which improves representation capabilities of the network. Experimentally, we comprehensively evaluate ChebyNet on diverse tasks, including function approximation, semantic segmentation, and visual recognition. Across all these tasks, ChebyNet consistently outperforms traditional neural networks under identical training conditions, demonstrating superior efficiency and fitting properties. Our findings underscore the potential of polynomial-based layer connections to significantly enhance neural network performance, offering a promising direction for future deep learning architectures.

## 1 Introduction

Deep Neural Networks (DNNs) have achieved remarkable progress across diverse areas (LeCun et al., 2015), including computer vision (Krizhevsky et al., 2012; He et al., 2016; Huang et al., 2017), natural language processing (Sutskever et al., 2014; Vaswani et al., 2017), reinforcement learning (Mnih et al., 2013), speech recognition (Hinton et al., 2012) and other fields (Dong et al., 2021; Wainberg et al., 2018). Despite these advancements, the underlying design paradigms often rely on a fixed layer structure where non-adjacent layers have limited interactions, typically restricted to additive shortcuts as seen in ResNets(He et al., 2016). Although considerable efforts have led to innovations such as dense connections(Huang et al., 2017), attention mechanisms(Vaswani et al., 2017), etc, the design paradigms constraint inherently restricts the expressive power and fitting capabilities of the network, thereby capping its potential performance.

The limited inter-layer relationships in neural networks constrain their learning and representational capabilities. Despite the introduction of additive shortcuts, these simplistic additive inter-layer connections still fail to provide the complex layer-wise interactions required(Bengio et al., 2013; Oyedotun et al., 2023). This observation raises a critical question: *how can we enhance the interaction between layers to improve both fitting capability and computational efficiency?*

In this paper, we propose ChebyNet, a novel architecture that leverages Chebyshev polynomial layer connections to enhance the representational capacity and efficiency of neural networks, motivated by the best uniform approximation properties and numerical stability of Chebyshev polynomials(Mason & Handscomb, 2002). Specifically, inspired by the approach of approximating numerical functions using a family of basis functions, we establish Chebyshev polynomial connections between layers in ChebyNet. The process involves two key steps: (1) generating Chebyshev basis functions of the layer features within the neural network, and (2) performing element-wise multiplication (Hadamard product) between these basis functions and the network's output at the current layer. The former establishes recursive relationship among adjacent layers while the latter constructs polynomial rela-

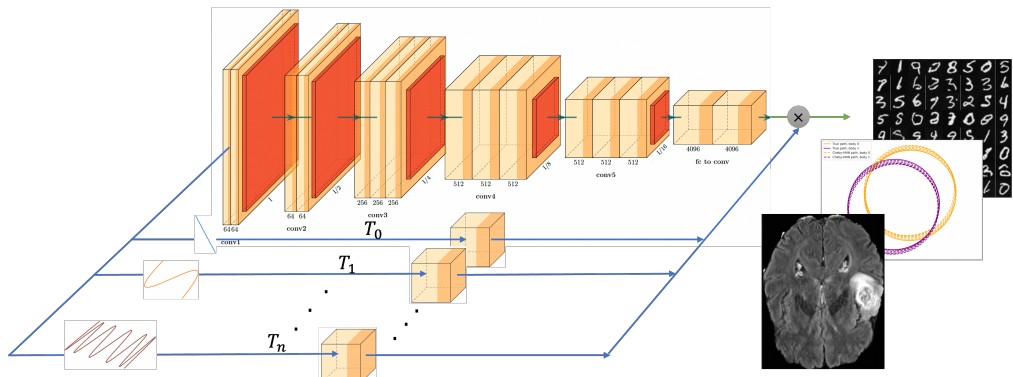

Figure 1: The workflow of the paper: Given a network, generating Chebyshev basis functions of the layer features within the neural network, and performing element-wise multiplication between these basis functions and the network's output at the current layer to establish Chebyshev polynomial connections between layers in ChebyNet. Subsequently, ChebyNet demonstrates potential for application across a diverse range of tasks, including function approximation, semantic segmentation, and image classification.

tionship between non-adjacent layers. Thoses operations yield a new set of features, whose weighted sum forms the network's output features.

By employing Chebyshev polynomial transformations, ChebyNet enables a more flexible and expressive representation of data that goes beyond simple additive shortcuts, allowing for the seamless integration of information from hierarchical layers, i.e., Chebyshev polynomial layers of different degrees. This approach not only enhances the fitting capabilities of the network but also optimizes its computational efficiency, addressing the growing concerns regarding resource utilization in deep learning models (Han et al., 2015).

We further evaluate the representational power and fitting capabilities of ChebyNet through a series of comprehensive experiments across various function approximation tasks, such as fitting numerical functions, image generation (fitting unknown functions or distributions), and physical law learning. We also validate the effectiveness of ChebyNet on classical computer vision tasks, i.e., semantic segmentation and visual recognition. Our results consistently demonstrate that ChebyNet outperforms traditional NNs under identical training conditions, highlighting its superior efficiency and fitting properties. Specifically, ChebyNet achieves better performance metrics with fewer parameters and reduced computational overhead, underscoring the practical benefits of polynomial-based layer connections.

Our main contributions are summarized below:

• We introduce ChebyNet, a novel neural network architecture that leverages Chebyshev polynomial connections to improve layer interactions, thereby enhancing the network's representational capacity and fitting accuracy.

• We provide comprehensive empirical evidence across a range of tasks, demonstrating that ChebyNet consistently surpasses existing network architectures in both task accuracy and computational efficiency.

• We highlight the potential of polynomial-based layer connections to substantially enhance neural network performance, presenting a promising avenue for future advancements in deep learning architectures.

Figure 1 illustrates the workflow and technical approach employed in this paper.

## 2 RELATED WORK

The success of DNNs relies on extensive research and thorough exploration of their architectures. Vast categories of DNN layers, such as the fully-connected layer (LeCun et al., 1998), convolutional

layer (Krizhevsky et al., 2012), pooling layer (Krizhevsky et al., 2012), batch normalization layer (Ioffe & Szegedy, 2015), etc, renders countless models like ResNet (He et al., 2016), gated recurrent unit (Cho et al., 2014), generative adversarial network (Goodfellow et al., 2014), Transformer (Vaswani et al., 2017), etc.

Despite the substantial progress in DNN design, the prevailing architectural paradigm often imposes limitations on the interaction between non-adjacent layers. Most conventional architectures employ fixed pathways for information flow, where connections between layers are primarily additive or sequential, restricting the model's ability to capture complex relationships within the data (He et al., 2016). Such designs restrict the approximation and representation capabilities of DNNs, making it difficult to learn intricate patterns that extend beyond local interactions. Consequently, there is a critical need for novel architectures that facilitate richer inter-layer connections, thereby enhancing the expressiveness and overall potential of neural networks.

Despite several studies, polynomial functions have been significantly underestimated in the construction of DNNs. According to Weierstrass's approximation theorem (Weierstrass, 1885; Stone, 1932), polynomial functions can approximate any continuous function with arbitrary precision, making them ideal candidates for activation functions in DNNs. Recent research has explored the integration of polynomial functions into DNNs, focusing on two primary aspects: (1) *polynomial activation functions (PAC)* and (2) *polynomial relationships between layers*. We elaborate on these approaches below.

Numerous attempts have been made to incorporate polynomial activation functions into neural networks. Following the ReLU construction methodology, (López-Rubio et al., 2019) introduced segmented polynomial activation functions, while (Loverich, 2015) demonstrated the superiority of these functions over segmented linear activation functions. However, the piecewise functions used in these experiments are non-differentiable, leading to an increased risk of overfitting during training. Additionally, the use of low-order polynomials reduces the network's nonlinearity, limiting its ability to learn and represent complex features. Optimization challenges also persist, with (Goyal et al., 2020) addressing these issues through the introduction of a new normalizing transformation.

Learnable parametric polynomial activation functions have also been proposed (Feng & Yang, 2023; Wu et al., 2018; Agostinelli et al., 2014; Piazza et al., 1993; Guarnieri et al., 1999), wherein the activation function parameters are learned during training or tuned via heuristic algorithms. However, these networks introduce higher computational complexity and pose difficulties in maintaining stable training conditions.

Orthogonal polynomials have been explored in activation function design due to their favorable mathematical properties. For instance, (Venkatappareddy et al., 2021) utilized Legendre polynomials for constructing activation functions, but (Deepthi et al., 2023) highlighted that Legendre polynomials may struggle to adapt to moderate and highly non-linear features. Similarly, Hermite polynomials have been employed in activation function design (Ma & Khorasani, 2005), but their effectiveness has been demonstrated only in networks with a single hidden layer. Chebyshev polynomials have also been employed as activation functions in several studies (Deepthi et al., 2023; Wang et al., 2022; Carini & Sicuranza, 2016; Sornam & Vanitha, 2018; Zhiqi, 2016; Lee & Jeng, 1998; Li et al., 2019), but unrestricted inputs may lead to unstable network training, necessitating normalization techniques (Wang et al., 2022; Li et al., 2019). Moreover, current research has solely validated Chebyshev polynomial activation functions in networks with a single hidden layer (Lee & Jeng, 1998).

Another promising direction involves establishing polynomial relationships between layers within the network. Previous work has primarily focused on low-order (typically second-order) polynomial connections, enabling quadratic interactions between layers (Chrysos et al., 2022). While this method has demonstrated some advantages in terms of improved representational capabilities, it remains limited in scope. Higher-order polynomial connections, particularly those with advantageous mathematical properties, have the potential to further enhance the network's expressiveness. However, these have not been extensively explored due to the significant computational and optimization challenges involved.

## 3 METHODOLOGY

In this section, we introduce ChebyNet, which is founded on two fundamental principles: the superior mathematical properties of Chebyshev polynomials and the innovative construction of polynomial relationships between layers.

### 3.1 CHEBYSHEV POLYNOMIALS

Chebyshev polynomials play a crucial role in approximation theory. The roots of Chebyshev polynomials of the first kind are employed in polynomial interpolation, producing polynomials that effectively mitigate the Runge phenomenon and offer optimal uniform approximation for continuous functions (Mason & Handscomb, 2002).

Chebyshev polynomials possess multiple definition formulations, such as trigonometric definition (as shown in equation 1), commuting polynomials definition, Pell equation definition, etc.

$$T_n(x) = \begin{cases} \cos(n \arccos x), & |x| \leq 1 \\ \cosh(n \operatorname{arccosh} x), & x > 1 \\ (-1)^n \cosh(n \operatorname{arccosh}(-x)), & x < -1 \end{cases} \tag{1}$$

Chebyshev polynomials can also be defined recursively, with the recursive relationship for Chebyshev polynomials of the first kind given by:

$$\begin{cases} T_0(x) = 1, \quad T_1(x) = x \\ T_{n+1}(x) = 2x T_n(x) - T_{n-1}(x), \quad n \geq 1 \end{cases} \tag{2}$$

The recursive property enables the stable generation of higher-order polynomials, which is beneficial for establishing stable and efficient connections between neural network layers.

Additionally, Chebyshev polynomials provide the tightest upper and lower bounds compared to all other polynomials on the interval [-1, 1], ensuring that the output remains constrained and does not diverge when used to construct DNNs. Chebyshev polynomials also offer the best uniform approximation to a continuous function under the maximum norm, enabling them to effectively capture complex patterns in data and enhance the network's representational power. These favorable mathematical properties endow Chebyshev polynomials with the feasibility for integration into network architectures.

### 3.2 CHEBYNET: ENHANCING LAYER INTERACTIONS WITH CHEBYSHEV POLYNOMIALS

Inspired by the approximation of numerical functions using a family of basis functions, ChebyNet incorporates Chebyshev polynomial connections to enhance interactions between network layers. The construction of ChebyNet centers around two key components: the recursive relationship among adjacent layers and the polynomial relationship between non-adjacent layers.

#### 3.2.1 RECURSIVE RELATIONSHIP AMONG ADJACENT LAYERS

The first aspect of ChebyNet's architecture is grounded in the recursive equation of Chebyshev polynomials, establishing a connection among three adjacent layers, as shown in equation 2. Specifically, given three consecutive layers, $L_{i-1}, L_i, L_{i+1}$, the output of layer $L_{i+1}$ is defined in terms of the outputs of the two preceding layers as follows:

$$L_{i+1}(x) = 2x \circ L_i(x) - L_{i-1}(x), \tag{3}$$

where $x$ represents the input features of the Chebyshev layer (also referred to as the intermediate representations within a DNN) and $\circ$ denotes the Hadamard product (element-wise multiplication). This recursive relationship facilitates stable and efficient propagation of information across the network layers.

It can be observed that this recursive structure among the three adjacent layers is, in fact, equivalent to constructing Chebyshev basis functions derived from the input features. By embedding the Chebyshev polynomial structure into the network topology, the model is able to learn complex relationships between layer outputs, significantly enhancing its representational and fitting capabilities.

### 3.2.2 POLYNOMIAL RELATIONSHIP BETWEEN NON-ADJACENT LAYERS

The second aspect involves establishing polynomial relationships between non-adjacent layers. This is achieved by performing element-wise multiplication between basis functions derived from the input features (or the output of the last layer) and the network's output at the current layer.

Assume that $x$ is the output of the previous layer, $f(\cdot)$ represents the transformation of the current layer, and $g(\cdot)$ denotes a down-sampling operation to align the dimensions of $x$ with the output $f(x)$ of the current layer. The aggregated features can then be expressed as follows in equation 4:

$$\text{Feature}_{agg} = f(x) \circ [L_0(g(x)) + L_1(g(x)) + \cdots + L_n(g(x))] = f(x) \circ \sum_{i=0}^{n} L_i(g(x)) \quad (4)$$

where $L_0(g(x))$ is an all-ones vector, meaning that $f(x) \circ L_0(g(x)) = f(x)$ represents the primary output of the current layer. This demonstrates that ChebyNet offers a novel extension and generalization of conventional neural network paradigms.

The combination of establishing recursive relationship among adjacent layers and aggregating polynomial relationship between non-adjacent layers effectively incorporates both local and global polynomial interactions within the network, significantly enhancing its representational power and fitting capabilities.

### 3.3 IMPLIMENTATION DETAILS

Although several closed-form expressions exist for calculating $n$-order Chebyshev polynomials, including trigonometric definitions (see equation 1), commuting polynomials, and Pell equation-based formulations, our extensive empirical studies demonstrate that the recursive formulation (see equation 2) provides superior numerical stability, which is crucial for ensuring consistent performance and facilitating reliable gradient propagation during the training process. This recursive approach effectively preserves the integrity of both the parameters and their corresponding gradients, significantly reducing the risk of overflow during computation. Considering the generally low-degree polynomials employed in practice, the additional computational complexity introduced by the recursive formulation is negligible and does not substantially affect the model's overall efficiency. Furthermore, we propose an optimized implementation leveraging dynamic programming principles, which strikes an efficient balance between time complexity ($O(n)$) and space complexity (($O(1)$)).

By definition, Chebyshev polynomials require their arguments to lie within the interval [-1, 1] by definition. Consequently, to effectively integrate Chebyshev polynomial-based layers into neural network architectures, it is essential to transform input vectors to meet this constraint. Our comprehensive empirical study reveals that applying non-linear transformations, such as Sigmoid and Softmax, to the input components substantially outperforms traditional linear normalization techniques.

## 4 EXPERIMENTS

To evaluate the effectiveness of ChebyNet, we conducted comprehensive experiments across three major tasks: function approximation, semantic segmentation, and image classification. The function approximation task encompasses fitting various numerical functions (approximating known functions), image generation (approximating unknown functions), and learning physical laws. Each experiment was designed to compare the performance of traditional neural network architectures with their ChebyNet counterparts.

### 4.1 FUNCTION APPROXIMATION

To assess the fitting capabilities of ChebyNet, we conduct a series of experiments focused on function approximation. This exploration highlights ChebyNet's effectiveness in approximating both known functions and unknown functions, as well as learning physical laws. The ability to generalize across these diverse domains is crucial for demonstrating the robustness of our model.

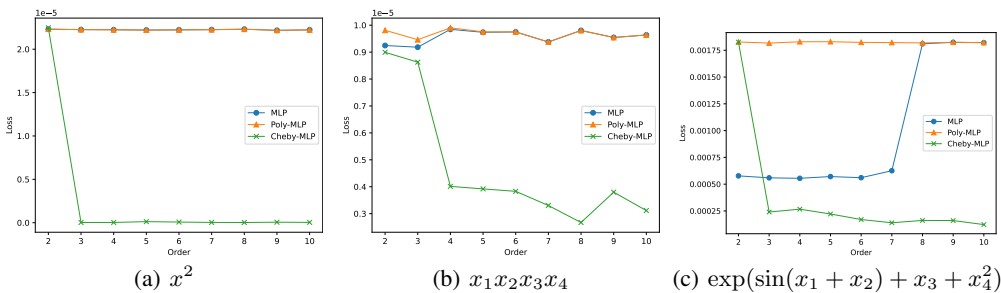

(a) $x^2$      (b) $x_1 x_2 x_3 x_4$      (c) $\exp(\sin(x_1 + x_2) + x_3 + x_4^2)$

Figure 2: The test MSE loss of the MLP and its ChebyNet and PolyNet variants with different polynomial orders on numerical functions. (a) $x^2$. (b) $x_1 x_2 x_3 x_4$. (c) $\exp(\sin(x_1 + x_2) + x_3 + x_4^2)$.

Table 1: The FID score of UNet-diffusion and Cheby-UNet-diffusion on MNIST.

| Order | 1 | 2 | 3 | 4 | 5 | 6 | 7 | 8 | 9 |
|---|---|---|---|---|---|---|---|---|---|
| MNIST ddpm with Baseline UNet-diffusion: 85.04 | | | | | | | | | |
| Cheby-UNet | 85.68 | **81.31** | **78.14** | **80.84** | 89.74 | 86.52 | **84.40** | **84.44** | 86.39 |

### 4.1.1 NUMERICAL FUNCTION APPROXIMATION (KNOWN FUNCTIONS)

We first utilize MLPs, along with their ChebyNet and PolyNet (substituting chebyshev polynomials in ChebyNet with ordinary polynomials) variants, to approximate a variety of numerical functions, ranging from simple elementary functions to more complex ones. Specifically, we evaluate the following univariate functions: $x^2$, $\sqrt{x}$, $\frac{1}{x}$, $\log x$, $\exp x$, $\sin x$, $\cos x$, $\arcsin x$, $\arccos x$, $\arctan x$, $\text{sgn} x$, $\text{sigmoid}(x)$, $\tanh x$, $\exp(-x^2)$, and the following multivariate functions of four variables: $x_1 x_2 x_3 x_4$, $\sin(x_1^2 + x_2^2 + x_3^2 + x_4^2)$, $\sin(x_1^2 x_2^2 x_3^2 x_4^2)$, $\exp(\sin(x_1 + x_2) + x_3 + x_4^2)$, $\exp(\sin(x_1^2 + x_2^2) + \sin(x_3^2 + x_4^2))$. The train and test datasets are generated from the selected functions, consisting of random inputs paired with their corresponding outputs. Each model, after being trained for 30 epochs, is evaluated using mean square error (MSE) loss on the test set. Several results are presented in Figure 2, with the remaining results shown in Figure 6 in Appendix A.

As shown in Figures 2 and 6, ChebyNet exhibits significantly lower test MSE loss compared to both the original network and PolyNet, highlighting its superior approximation capabilities relative to the baseline. Additionally, ChebyNet's test loss decreases as the polynomial order increases, indicating that higher-order approximations lead to enhanced fitting capacity.

### 4.1.2 IMAGE GENERATION (UNKNOWN FUNCTIONS)

Image generation of diffusion models can be conceptualized as a two-step process (Ho et al., 2020): first, approximating a target distribution or function in the latent space, and then sampling from this distribution to generate images. Consequently, the quality of the generated images reflects the model's ability to accurately fit the target function.

We employed UNet-diffusion (Ho et al., 2020) and its ChebyNet variant (Cheby-UNet-diffusion) to generate images based on the MNIST dataset. Each model was trained for 1,000 epochs with a batch size of 64. We conducted 5 rounds of training and sampling, with each round involving 1,000 sampling steps to generate 1,000 images. The quality of the generated images was assessed using the Frechet Inception Distance (FID) score (Heusel et al., 2017) in Table 1. As shown in Table 1, Cheby-UNet-diffusion achieves lower FID scores than the baseline across most polynomial orders, indicating that ChebyNet has an enhanced capacity for learning unknown functions or distributions. We also visualize 64 generated samples for comparison, as shown in Figure 3.

(a) UNet-diffusion        (b) Cheby-UNet-diffusion

Figure 3: The sampling images of UNet-diffusion and Cheby-UNet-diffusion on MNIST. (a) UNet-diffusion. (b) Cheby-UNet-diffusion.

Table 2: The mse-loss between simulated trajectory and ground truth predicted by NODE, HNN and their ChebyNet variants.

| Order | 1 | 2 | 3 | 4 | 5 | 6 | 7 | 8 | 9 |
|---|---|---|---|---|---|---|---|---|---|
| bouncing ball with Baseline NODE: 0.231 | | | | | | | | | |
| Cheby-NODE | 0.499 | 0.286 | **0.225** | **0.020** | **0.071** | 0.376 | **0.024** | 0.451 | **0.084** |
| 2-body problem with Baseline HNN: 6.404 (Unit: 1e-6) | | | | | | | | | |
| Cheby-HNN | **4.474** | **5.994** | **2.437** | **2.220** | 6.545 | **5.727** | **4.245** | **4.454** | **3.087** |
| 3-body problem with Baseline HNN: 4.437 (Unit: 1e-1) | | | | | | | | | |
| Cheby-HNN | 4.981 | 4.531 | **4.242** | **4.373** | **4.121** | **4.428** | **4.029** | 4.513 | **4.219** |
| real pendulum problem with Baseline HNN: 5.982 (Unit: 1e-3) | | | | | | | | | |
| Cheby-HNN | **5.807** | **5.794** | **5.805** | **5.808** | **5.803** | **5.802** | **5.793** | **5.806** | **5.807** |

### 4.1.3 PHYSICAL LAW LEARNING

While previous results have demonstrated ChebyNet's robust fitting capabilities for both known and unknown functions, concerns about potential overfitting naturally arise. To address this, we conduct experiments on modeling object trajectories in real-world physical scenarios. To be concrete, we employ Neural ODEs (NODE)(Chen et al., 2018) and Hamiltonian Neural Networks (HNN)(Greydanus et al., 2019) which are commonly used for such scenarios, and compare the kinetic energy, potential energy, and total mechanical energy of the objects between the baselines and the ChebyNet variants (Cheby-NODE and Cheby-HNN). We use NODE and Cheby-NODE to model the trajectory of a bouncing ball, training for 1,000 iterations, and employ HNN and Cheby-HNN to simulate two-body and three-body problems, as well as the motion of a real pendulum, with 10,000 iterations of training. For HNN and Cheby-HNN, we measure both the trajectory errors and energy discrepancies. Overfitting would manifest as small trajectory errors coupled with large energy discrepancies.

The trajectories of the 2-body problem predicted by HNN and Cheby-HNN are shown in Figure 4, while additional results are presented in Appendix B due to space limitations. These include: (1)

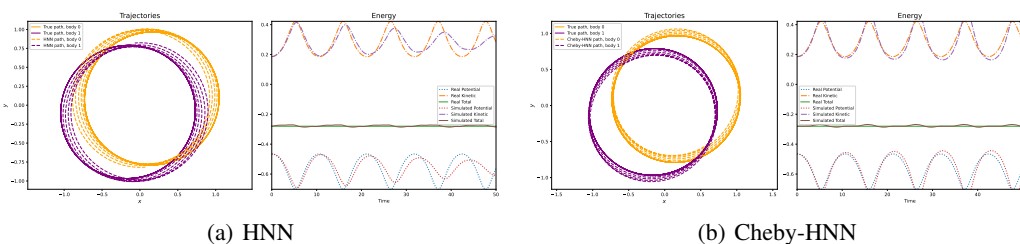

(a) HNN                  (b) Cheby-HNN

Figure 4: The 2-body trajectories predicted by HNN and Cheby-HNN. (a) HNN. (b) Cheby-HNN.

the trajectories of the 3-body problem and a real pendulum predicted by HNN and Cheby-HNN in Figure 8 and Figure 10, respectively; (2) energy predictions for the 2-body and 3-body problems across various seeds, using HNN and Cheby-HNN, in Figure 7 and Figure 9; and (3) the trajectories of a bouncing ball predicted by NODE and Cheby-NODE in Figure 11.

The mse-loss of energy of 2-body, 3-body, and real pendulum problem of HNN and Cheby-HNN is shown in Table 3:

Table 3: The mse-loss of energy of various physical scenarios of HNN and Cheby-HNN.

| Scenario | 2-body | 3-body | real pendulum |
|---|---|---|---|
| HNN | $2.903 \times 10^{-5}$ | $1.096 \times 10^{-2}$ | $7.500 \times 10^{-3}$ |
| Cheby-HNN | $1.085 \times 10^{-5}$ | $6.093 \times 10^{-3}$ | $7.494 \times 10^{-3}$ |

Through these comprehensive experiments, we have demonstrated that ChebyNet exhibits superior approximation capabilities compared to the original network architecture, while showing no signs of overfitting. This robust performance suggests that ChebyNet is well-suited for learning and modeling physical laws.

### 4.2 SEMANTIC SEGMENTATION

We further demonstrate that ChebyNet is capable of enhancing performance on classical computer vision tasks. We conduct a series of experiments focused on semantic segmentation, a challenging task that requires precise delineation of objects within an image. This task is particularly relevant in applications such as medical diagnosis, autonomous driving, and scene understanding, where accurate pixel-level classification is crucial.

We evaluate the effectiveness of Chebyshev polynomial connections based on the classical UNet architecture. UNet, recognized for its unique structure, employs skip connections that enable direct interactions between layers at varying depths (Ronneberger et al., 2015). These connections allow the model to combine high-level semantic information with low-level spatial details, improving its ability to generate precise segmentation masks. However, while UNet establishes these interactions, they remain relatively straightforward. In contrast, Cheby-UNet, the ChebyNet variant of Unet, aims to leverage Chebyshev polynomial connections to facilitate more complex and nuanced interactions across layers.

We compare the segmentation performance on two prominent datasets: ACDC (Automated Cardiac Diagnosis Challenge)(Bernard et al., 2018) and BraTS19 (Brain Tumor Segmentation Challenge)(Bakas, 2020). To thoroughly assess performance across different settings, we implement a series of configurations, including 2D fully supervised, 2D semi-supervised, and 3D fully supervised experiments. The Dice scores of both UNet and Cheby-UNet are presented in Tab. 4. We randomly select an image from the test set for visualization, with the results depicted in Fig. 12 in Appendix C.

The results of our experiments demonstrate that Cheby-UNet consistently outperforms the traditional UNet architecture across all experimental configurations, as measured by the Dice metric.

Table 4: The dice value of UNet and Cheby-UNet on ACDC and BraTS19.

| Order | 1 | 2 | 3 | 4 | 5 | 6 | 7 | 8 | 9 |
|---|---|---|---|---|---|---|---|---|---|
| ACDC 2D-fully-supervise with Baseline UNet: 0.7984 | | | | | | | | | |
| Cheby-UNet | **0.8122** | 0.7900 | **0.8070** | **0.8031** | 0.7923 | **0.8040** | **0.8077** | 0.7896 | **0.8013** |
| ACDC 2D-semi-supervise with Baseline UNet: 0.8225 | | | | | | | | | |
| Cheby-UNet | **0.8305** | **0.8270** | 0.8205 | **0.8320** | **0.8328** | **0.8243** | **0.8338** | **0.8328** | **0.8339** |
| BraTS19 3D-fully-supervise with Baseline UNet: 0.8291 | | | | | | | | | |
| Cheby-UNet | **0.8306** | **0.8389** | **0.8415** | **0.8448** | **0.8404** | 0.8279 | **0.8324** | **0.8417** | **0.8365** |

Table 5: The test accuracy of models and their ChebyNet on CIFAR100.

| Order | 1 | 2 | 3 | 4 | 5 | 6 | 7 | 8 | 9 |
|---|---|---|---|---|---|---|---|---|---|
| Using PCNN Architecture with Baseline: 59.8 | | | | | | | | | |
| Poly-PCNN | **59.8** | 59.5 | 59.4 | 59.4 | 59.8 | 60.0 | 59.7 | 59.7 | 59.6 |
| Cheby-PCNN | 59.7 | **60.5** | **60.2** | **59.9** | **60.3** | **60.4** | **60.4** | **60.0** | **60.2** |
| Using MobileNet Architecture with Baseline: 60.0 | | | | | | | | | |
| Poly-MobileNet | 59.7 | 59.8 | 59.3 | 60.1 | 60.2 | **60.8** | 59.6 | **60.1** | O |
| Cheby-MobileNet | **60.0** | **60.0** | **60.2** | **60.5** | **60.4** | 60.4 | **60.3** | 60.0 | **60.2** |
| Using ResNet18 Architecture with Baseline: 76.1 | | | | | | | | | |
| Poly-ResNet18 | 75.6 | **76.7** | 76.0 | 76.4 | **76.4** | 76.0 | 75.8 | 75.9 | 75.7 |
| Cheby-ResNet18 | 75.8 | 75.5 | **76.3** | **76.6** | 76.1 | **76.6** | **76.4** | **76.3** | **76.1** |
| Using ResNet34 Architecture with Baseline: 76.5 | | | | | | | | | |
| Poly-ResNet34 | **76.7** | **77.0** | **77.3** | 76.2 | **76.8** | **76.9** | **76.7** | **76.7** | 76.2 |
| Cheby-ResNet34 | 76.5 | **77.0** | 77.1 | **76.7** | **76.8** | 76.8 | 75.8 | **76.7** | **76.6** |

This improvement highlights the effectiveness of ChebyNet in enhancing the model's representational power, enabling better feature extraction and segmentation accuracy. Furthermore, the results reinforce our hypothesis regarding the utility of complex inter-layer relationships. By integrating Chebyshev polynomial connections, Cheby-UNet effectively captures intricate interactions between layers, leading to superior performance in delineating object boundaries and recognizing subtle distinctions between classes.

## 4.3 IMAGE CLASSIFICATION

We further evaluate the performance of ChebyNet on the classification task using the CIFAR-10 (Krizhevsky et al., 2009) and CIFAR-100 (Krizhevsky et al., 2009) datasets. The baseline models, including MLP, PCNN (a plain 5-layer CNN with 5 hidden states), MobileNetV2, ResNet18, and ResNet34, and the corresponding ChebyNet and PolyNet variants, are trained from scratch for 120 epochs. The batch size for each model is set to 128, with an initial learning rate of 0.1, which is reduced by a factor of 10 at epochs 40, 60, 80, and 100. We use SGD with momentum of 0.9 and a weight decay of $5 \times 10^{-4}$ as the optimizer. All other settings for both the baseline models and ChebyNet are kept identical. The test accuracy of various models on CIFAR-10 and CIFAR-100 are presented in Table 6 (in Appendix D), Table 7 (in Appendix D) and Table 5, respectively, where 'O' in the tables stands for numerical overflow.

Through these comprehensive experiments, we have demonstrated that ChebyNets of most orders perform better than the original models. Besides, the recursive formulation of Chebyshev polynomials avoid numerical overflow. We can conclude that the learning and optimization ability of ChebyNet is exceeding that of the corresponding model and PolyNet thus it presents a promising approach for application in various visual tasks.

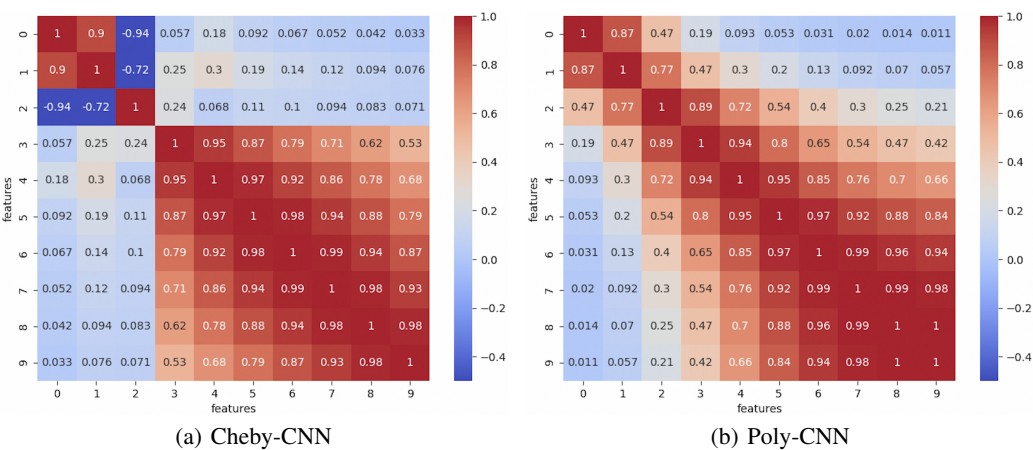

Figure 5: The cosine similarity among features of different orders. (a) Cheby-CNN. (b) Poly-CNN.

To elucidate the irregular performance variations of the same model across different approximation orders, we examine this phenomenon by computing the correlation matrices between features of different orders for both Cheby-CNN and Poly-CNN at the final epoch of training, as shown in Figure 5. Notably, Cheby-CNN exhibits lower overall feature similarity compared to Poly-CNN. In particular, low-order features (the first three orders) display weak or even negative correlations, while higher-order features demonstrate stronger correlations. This reveals several advantageous properties of Cheby-CNN:

• The element-wise multiplication with orthogonal polynomials reduces feature correlation, facilitating the extraction of more compact data structures.

• The strong correlation among high-order features suggests that low-order features are already sufficient for representing the underlying information, indicating potential for parameter compression.

## 5 CONCLUSION

In this paper, we propose ChebyNet, a novel neural network architecture that utilizes Chebyshev polynomial connections to enhance representational capacity and computational efficiency. By introducing polynomial transformations between layers, ChebyNet offers a more expressive and flexible framework with robust approximation capabilities, enabling superior performance in various tasks. Our empirical results demonstrate that ChebyNet consistently outperforms traditional neural networks in terms of both accuracy and resource efficiency, making it a promising approach for advancing deep learning models. These findings suggest that polynomial-based layer connections could play a key role in future neural network developments.

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

## A APPENDIX 1: NUMERICAL FUNCTION APPROXIMATION

This appendix presents supplementary results from the numerical approximation experiments described in Section 4.1.1.

The test mse-loss of various numerical functions in shown in Figure 6:

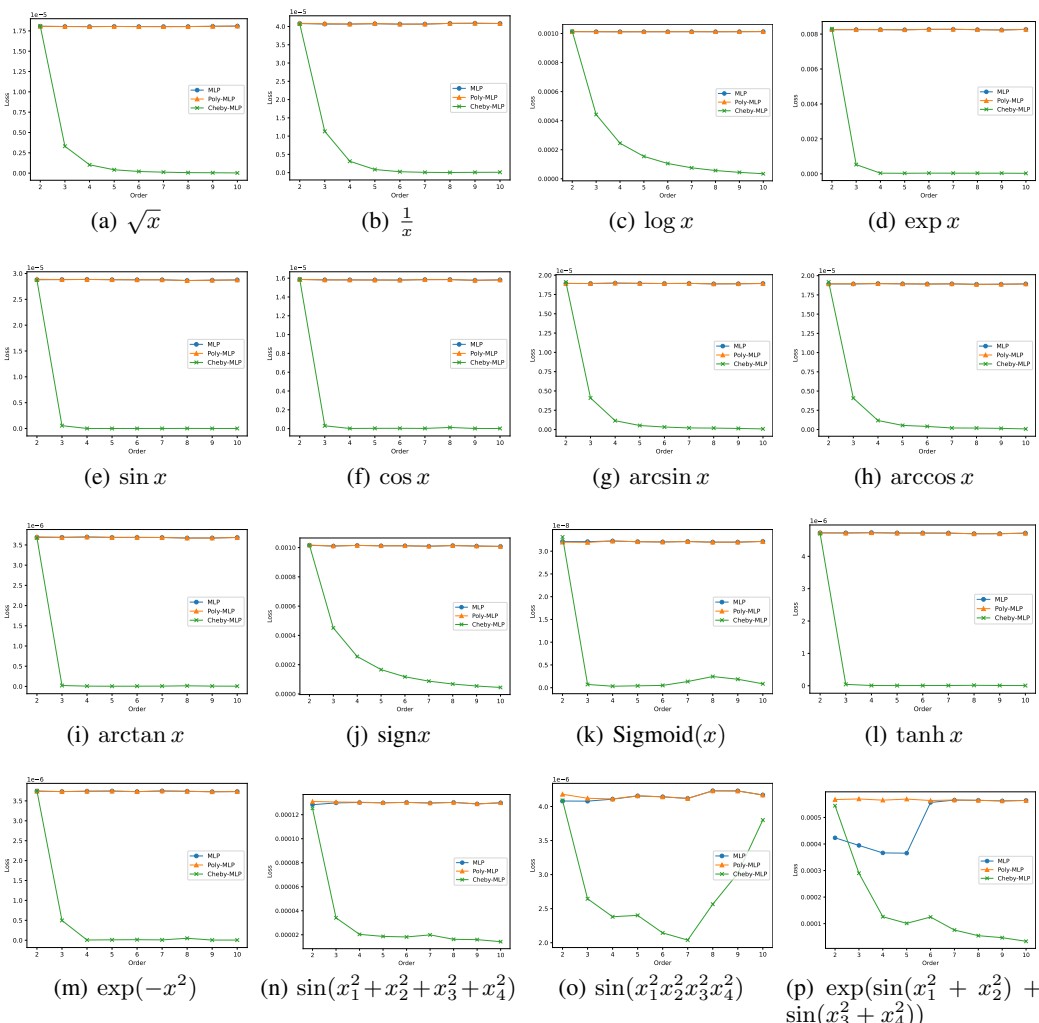

Figure 6: The test MSE loss of MLP and its ChebyNet and PolyNet variants of different orders on various numerical functions.

# B APPENDIX 2: PHYSICAL LAW LEARNING

This appendix presents supplementary results from the physical law learning experiments described in Section 4.1.3.

The energy of 2-body problem predicted by HNN and Cheby-HNN across various seeds are shown in Figure 7:

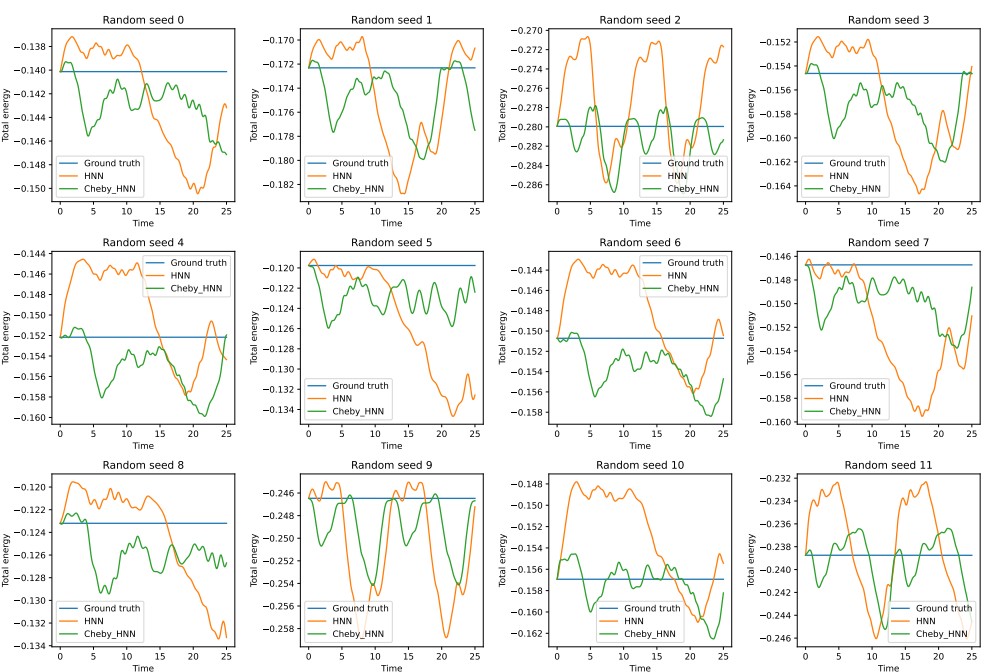

Figure 7: The 2-body energy predicted by HNN and Cheby-HNN across various seeds. (a) HNN. (b) Cheby-HNN.

The trajectories of 3-body problem predicted by HNN and Cheby-HNN are shown in Figure 8:

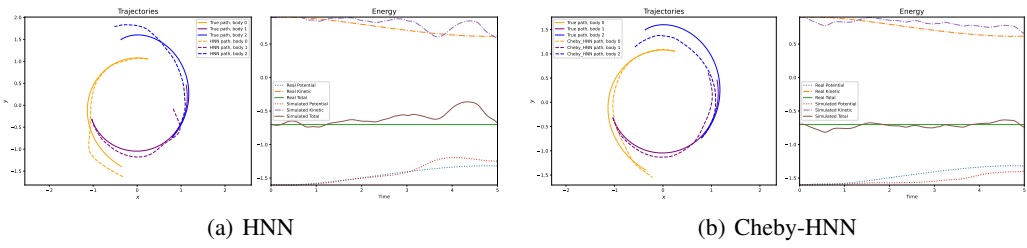

(a) HNN        (b) Cheby-HNN

Figure 8: The 3-body trajectories predicted by HNN and Cheby-HNN. (a) HNN. (b) Cheby-HNN.

The energy of 3-body problem of HNN and Cheby-HNN across various seeds are shown in Figure 9:

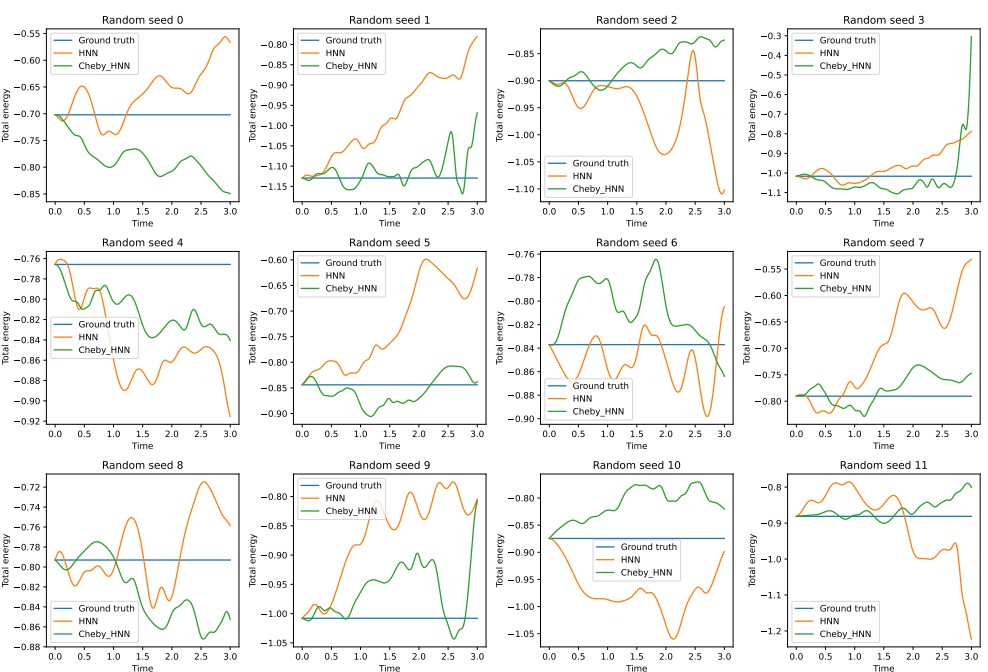

Figure 9: The 3-body energy predicted by HNN and Cheby-HNN across various seeds. (a) HNN. (b) Cheby-HNN.

The trajectories of a real pendulum predicted by HNN and Cheby-HNN are shown in Figure 10:

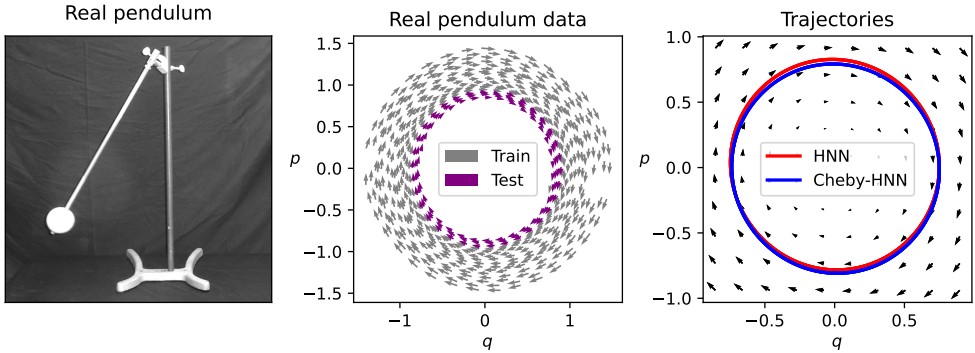

Figure 10: The real pendulum trajectories predicted by HNN and Cheby-HNN. (a) HNN. (b) Cheby-HNN.

The trajectories of a bouncing ball predicted by NODE and Cheby-NODE are shown in Figure 11:

## C  APPENDIX 3: SEMANTIC SEGMENTATION

This appendix presents supplementary results from the semantic segmentation experiments described in Section 4.2.

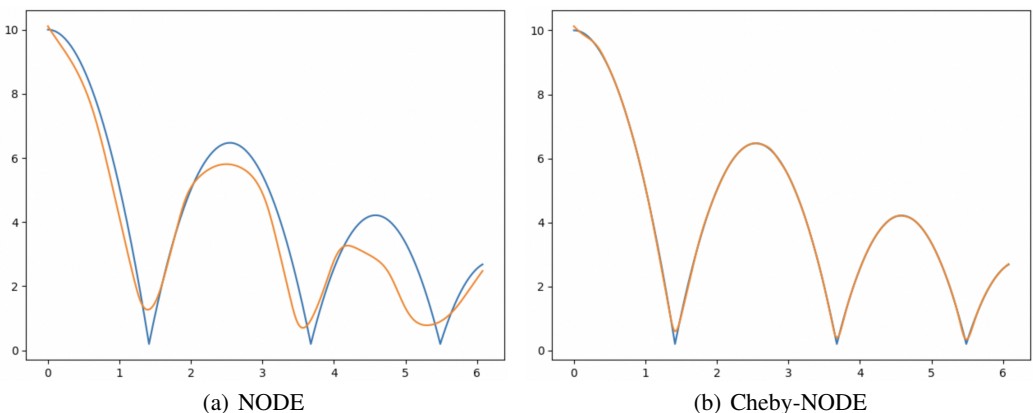

(a) NODE

(b) Cheby-NODE

Figure 11: The trajectories of a bouncing ball predicted by NODE and Cheby-NODE. (a) NODE. (b) Cheby-NODE.

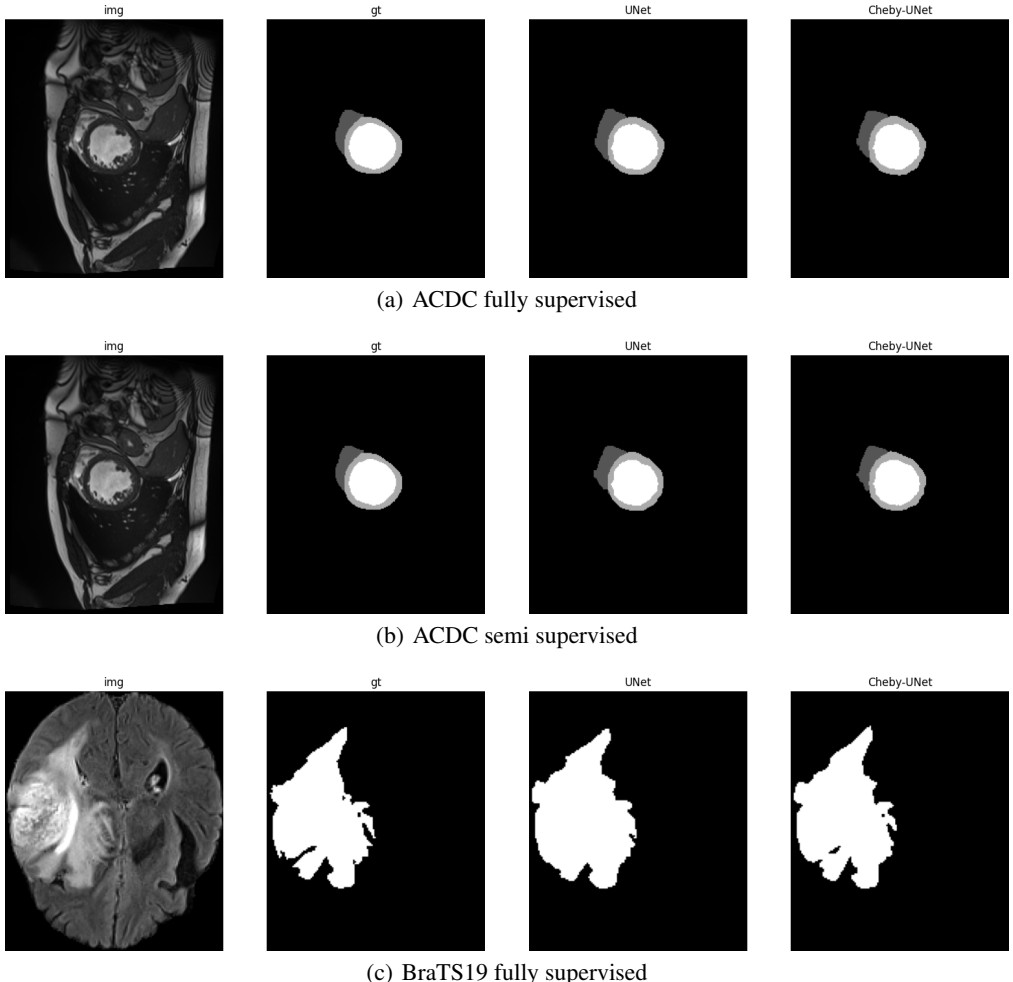

(a) ACDC fully supervised

(b) ACDC semi supervised

(c) BraTS19 fully supervised

Figure 12: The segmentation results of ACDC and BraTS19. (a) ACDC for fully supervised task. (b) ACDC for semi supervised task. (c) BraTS19 for fully supervised task.

## D  APPENDIX 4: IMAGE CLASSIFICATION

This appendix presents supplementary results from the image classification experiments described in Section 4.3.

The test accuracy of MLP models on CIFAR-10 and CIFAR-100 are presented in Table 6, while that of other models on CIFAR-10 in Table 7.

Table 6: The test accuracy of models and their ChebyNet on CIFAR using MLP as the backbone.

| Order | 1 | 2 | 3 | 4 | 5 | 6 |
|---|---|---|---|---|---|---|
| Experimental Results on CIFAR-10 | | | | | | |
| MLP | **55.0** | **55.2** | 54.5 | 54.2 | 53.5 | 53.3 |
| Poly-MLP | 53.6 | 54.1 | **55.1** | O | O | O |
| Cheby-MLP | 53.6 | 53.9 | 54.7 | **55.8** | **56.5** | **56.2** |
| Experimental Results on CIFAR-100 | | | | | | |
| MLP | 26.5 | 26.2 | 27.0 | 25.6 | 25.7 | 25.7 |
| Poly-MLP | 27.5 | **27.9** | 27.5 | 27.1 | O | O |
| Cheby-MLP | **27.5** | 27.7 | **27.8** | **29.8** | **29.8** | **30.1** |

Table 7: The test accuracy of models and their ChebyNet on CIFAR10.

| Order | 1 | 2 | 3 | 4 | 5 | 6 | 7 | 8 | 9 |
|---|---|---|---|---|---|---|---|---|---|
| Using PCNN Architecture with Baseline: 87.2 | | | | | | | | | |
| Poly-PCNN | 87.3 | 87.3 | 87.4 | 87.7 | **87.6** | 87.5 | **88.0** | 87.2 | 87.2 |
| Cheby-PCNN | **87.4** | **87.7** | **87.6** | **87.8** | 87.4 | **87.6** | 87.7 | **87.7** | **87.6** |
| Using MobileNet Architecture with Baseline: 84.7 | | | | | | | | | |
| Poly-MobileNet | **85.0** | 84.6 | 84.8 | 84.8 | **84.9** | 84.9 | 84.4 | 85.0 | 84.5 |
| Cheby-MobileNet | 84.9 | **85.1** | **84.8** | **85.1** | **84.9** | **85.0** | **85.3** | **85.4** | **84.6** |
| Using ResNet-18 Architecture with Baseline: 94.5 | | | | | | | | | |
| Poly-ResNet18 | **94.6** | 94.6 | 94.4 | 94.3 | **94.7** | 94.3 | 94.3 | 94.3 | 94.4 |
| Cheby-ResNet18 | 94.3 | **94.7** | 94.4 | **94.5** | 94.3 | **94.6** | 94.4 | 94.4 | **94.6** |
| Using ResNet-34 Architecture with Baseline: 94.6 | | | | | | | | | |
| Poly-ResNet34 | 94.6 | 94.4 | 94.6 | 94.7 | **94.7** | 94.6 | 94.7 | 94.3 | **94.8** |
| Cheby-ResNet34 | **94.8** | 94.4 | **94.8** | **94.8** | 94.3 | **94.7** | **94.8** | **94.6** | **94.6** |

