# OpenReview forum: "ChebyNet: Boosting Neural Network Fitting and Efficiency through Chebyshev Polynomial Layer Connections"
_ICLR.cc/2025/Conference — ICLR 2025 Conference Withdrawn Submission_

### Official Review · Reviewer_f6mi · 2024-10-26

**Soundness:** 2
**Presentation:** 1
**Contribution:** 3
**Rating:** 3
**Confidence:** 4

**Summary:**

The paper proposes ChebyNet, a neural network architecture that uses Chebyshev polynomial connections to boost the network's fitting capabilities and efficiency. The idea is to go beyond typical additive shortcuts, adding both recursive connections between adjacent layers and polynomial-based relationships between non-adjacent layers. The authors demonstrate the effectiveness of ChebyNet on various tasks, like function approximation, image classification, and semantic segmentation, showing that it often outperforms standard networks with fewer parameters.

**Strengths:**

1. Novel Use of Polynomials: Applying Chebyshev polynomial connections for inter-layer relationships is an interesting twist that brings more flexibility to the model's structure.
2. Versatility Across Tasks: The method shows improvements across different tasks, suggesting it has general applicability.

**Weaknesses:**

1. Implementation: The implementation details are unclear. In the methodology section, Equations 3 and 4 outline the connectivity patterns between layers, but there is no specific guidance on how to apply these connections to complex architectures like UNet (as shown in Tables 1 and 4), or ResNet and MobileNet (as shown in Table 5). This raise serious problems when I want to to dive in the details of the paper. Those details are also not incorporated in the Appendix as well.

2. Unclear Use of Polynomials: The method appears to focus on a recursive layer connectivity similar to Chebyshev polynomials, but it doesn't actually involve using polynomials. Equation 4 resembles Equation 3 but starts with a different initial condition, leading to entirely different sequences in the recursion.

3. Computation: While the paper claims the efficiency (Line 88-89, "fewer parameters and reduced computational overhead"), there is no actually discussion on the real computation gain with respect to different applications. From my understanding, with increased connectivity, there is a likelihood of higher computational costs, which is why architectures like DenseNet, despite their strong performance, are not widely adopted in real-world applications. The paper does not sufficiently discuss how ChebyNet handles the potential slowdown due to the additional polynomial connections.

4. Limited Baseline Comparisons: The paper mainly introduce a new type of connectivity of layers, which is more on par for ResNet and DenseNet. However, the comparisons are mostly against basic versions of popular models. Adding comparisons with more sophisticated connectivity strategies would strengthen the results and make the findings more convincing.

**Questions:**

- How do you integrate the Chebyshev connections into complex architectures like ResNet or UNet? Can you provide more concrete details?

- Given that Equation 4 diverges from the typical polynomial sequence, what justifies calling the method polynomial-based? Is the benefit truly coming from the polynomial structure or just from additional learned connections?

---

### Official Review · Reviewer_F5nM · 2024-10-29

**Soundness:** 1
**Presentation:** 2
**Contribution:** 2
**Rating:** 3
**Confidence:** 4

**Summary:**

Current architectures used in the field of deep learning are limited in the modeling of relationships between non-adjacent layers. Skip-connections are popular additive methods for connecting non-adjacent layers, and prior work has explored the use of polynomial functions to establish relationships between layers. In this work, Chebyshev polynomials of high order have been applied to several modern architectures and evaluated on several tasks. The experiments show that adding Chebyshev polynomials to the architectures can help improve performance slightly when compared with certain baselines.

**Strengths:**

Originality
This work explores an under-explored direction of research. Prior work has not carried out such an extended study.

Quality
The method has been applied to many different tasks, trying to show the potential applications to several areas where deep learning is traditionally applied.

Significance
Specific problems will require specific biases, and Chebyshev polynomials are indeed an interesting way to provide useful modeling biases to the available architectures.

**Weaknesses:**

The contribution is not clear. The first contribution claimed states that ChebyNet is introduced. However, several architectures are apparently used in the experiments, where Chebyshev polynomials are somehow applied to existing architectures to attempt improvements in their performance. Please clarify the first contribution. The second contribution claims that ChebyNet consistently surpasses existing networks, however, in the experiments it is clear that this is only true for certain hyperparameter choices (which are not clear).

The methodology is far from being clear or reproducible. The polynomials are described, but how they are applied to the network architecture is never explained in sufficient detail to allow an expert to reproduce the results obtained. No details on the architecture structure, no pseudocode of the implementation, no details of the optimizer used for each experiment (it is mentioned only for one) or the learning rate, weight decay, or other details on data augmentation, and so on.

There are writing issues with the manuscript, please read it over again and fix grammatical and typographical mistakes (e.g. Implimentation as the title of section 3.3).

Numerical function approximation loss un Figure 2 shows Loss against Order. What does Order mean for an MLP? Please, again, do not place results and experiments without explaining what was done. Why is the MLP failing to fit a quadratic function? Was the error achieved exactly 0? This might not be surprising given that polynomials are part of the architecture itself, but would have been interesting a more in-depth discussion.

The FID obtained on MNIST appears very high for both the Cheby-UNet, and the baseline UNet. More details on the hyperparameters used would help understand the performance. The quality of the samples also appears qualitatively worse than a simple UNet-based implementation of diffusion available on GitHub (https://github.com/bot66/MNISTDiffusion). More details are necessary.
The use of FID as a metric is not sufficient. As the objective is showing the ability of the architecture to fit complex functions, the log-likelihood would have also been an important metric to display, as it more closely shows the ability of a model to fit complex functions. MNIST is not enough, and at least CIFAR10 should have been used. I would also suggest CelebA, which appears more complex but is actually quite simple compared to CIFAR10 for a generative model.

There was no discussion or acknowledgment of the limitation which comes from using models with different parameter count. From the text it is not clear whether the parameters count was kept constant, or if at least the comparison could be considered fair in all experiments.

**Questions:**

Clear method section, with a straightforward explanation of the architectures used would go a long way in understanding the significance of the experiments.

What was the reason behind the choice of experiments and baseline architectures?

Why are Chebishev polynomials particularly good? This is not really clear from the text. Are they a fundamental ingredient that all modern architectures should use to propel their performance further?

Could you run some experiments regarding the precise type of polynomials used, or more clear ablations on where and how the polynomials were applied?

---

### Official Review · Reviewer_rj4R · 2024-11-03

**Soundness:** 3
**Presentation:** 3
**Contribution:** 2
**Rating:** 5
**Confidence:** 3

**Summary:**

In this paper, the author addresses the issue that previous neural network architectures fail to explicitly model the relationships between different layers. To mitigate this, the author introduces ChebyNet, which models the recursive and polynomial relationships between layers. To validate the proposed method, the authors conduct several experiments across various tasks. The results demonstrate that incorporating relationships between layers enhances performance and suggests a promising direction for network structural design.

**Strengths:**

This paper is well-crafted, effectively illustrating the concepts and experimental results.
To validate the proposed method, the authors conduct extensive experiments across various tasks, including image classification and image segmentation. The outcomes of these experiments confirm the efficacy of the proposed method.

**Weaknesses:**

Despite the demonstrated experimental improvements, I have several concerns regarding the proposed method. Firstly, could the authors provide an analysis of the memory usage, inference time, and training time of the proposed method? I am interested in determining whether it requires additional resources to train the model. Additionally, the use of MNIST and CIFAR datasets might not be sufficient to thoroughly validate the method; could the authors present results on larger datasets? Furthermore, could the authors discuss the robustness of the proposed method? While modeling the relationship between different layers may increase the capacity of the model, it could also increase the risk of overfitting.

**Questions:**

please refer to weakness.

---

### Official Review · Reviewer_ryV2 · 2024-11-04

**Soundness:** 2
**Presentation:** 2
**Contribution:** 2
**Rating:** 3
**Confidence:** 3

**Summary:**

This paper introduces ChebyNet, a novel architecture aimed at enhancing neural networks by fostering connections between non-adjacent layers, an area typically underexplored in conventional networks. The core motivation is the limited interaction between distant layers, which can constrain a network's capacity to model complex functions. ChebyNet addresses this by employing Chebyshev polynomial basis functions to augment layer connections, which are then fused with outputs, effectively enhancing the network’s representational power. The proposed method is evaluated across various tasks, including regression, image generation on MNIST, and classification, demonstrating that ChebyNet is versatile and improves performance in numerous settings.

**Strengths:**

1.	ChebyNet is adaptable and can be seamlessly integrated into multiple existing architectures, such as UNet and HNN, with minimal implementation complexity.
2.	The approach is tested on a range of tasks and exhibits performance gains in most cases, supporting its practical efficacy.
3.	ChebyNet shows a robust capacity for approximating various mathematical functions, with Table 6 in the appendix indicating superior performance over MLP and Poly-MLP in approximating elementary functions like sign(x) and tanh.

**Weaknesses:**

1.	The motivation for ChebyNet could be further clarified. While the paper states that existing networks lack inter-layer connections, there are established models, like DenseNet, that enhance layer interactions. Thus, the benefit of Chebyshev polynomial-based connections versus simpler dense, residual or pyramid connections remains unclear.
2.	Figure 1 could be refined for clarity, as it currently suggests that polynomial connections link the network’s input and output directly, whereas, according to the text part, these connections are applied within layers.
3.	The method’s evaluation is limited to small-scale datasets. Testing on larger benchmarks, such as ImageNet, would provide a more compelling demonstration of its scalability.
4.	While ChebyNet is posited to improve non-adjacent layer interactions, the paper lacks strong empirical / theoretical evidence to substantiate this claim fully.
5.	The paper claims that Chebyshev connections can enhance the efficiency of DNNs; however, no experiments are provided to validate this claim. To my knowledge, additional connections may introduce extra memory and I/O overhead during inference. Supplementary experiments demonstrating the efficiency benefits would strengthen the paper."

**Questions:**

Please refer to the weaknesses, particularly the motivation for using Chebyshev polynomials to enhance DNNs. How does this method compare to simpler residual or dense connections in terms of inter-layer interaction benefits?

---

### Official Review · Reviewer_6oyp · 2024-11-07

**Soundness:** 3
**Presentation:** 3
**Contribution:** 2
**Rating:** 5
**Confidence:** 4

**Summary:**

The paper studies a new approach for interactions between non-adjacent layers in neural networks. Existing interactions among non-adjacent layers are typically studied as additive shortcuts as ResNets, dense connections, and attention mechanisms. This paper aims to bring a new type of interactions between nonadjacent network layers, by multiplying Chebyshev polynomials of inputs (up to a downsampling) to features as element-wise or Hadamard products. Since the 0-order Chebyshev polynomial is the identity, such a scheme can be regarded as an extension of existing network features $f(x)$ to those multiplied element-wisely by sums of high order Chebyshev polynomials of inputs, i.e. $f(x) \circ \sum_{i=0}^n L_i(g(x))$, where $L_i$ is defined as the Chebyshev polynomials of the first kind recursively and $g(x)$ is a downsampling operation to align the dimensionality of input $x$ with the feature $f(x)$.

The motivation of exploiting Chebyshev polynomials roughly lies in the fact that their roots, Chebyshev nodes, actually provide a tight bound in polynomial interpolation of continuous functions that minimizes the Runge oscillation phenomenon. Moreover, Chebyshev polynomials in the first kind has a recursive representation which can be easily implemented with deep neural networks.

The utility of such a construction is demonstrated by several experiments: low dimensional numerical function approximation, MNIST image generation using UNet-diffusion, learning of some dynamical systems (2-body, 3-body, and real pendulum problem), UNet image segmentation (ACDC and BraTS19), and image classification (Cifar-10 and Cifar-100).

**Strengths:**

1) The idea of multiplying Chebyshev polynomials of inputs to features as element-wise (Hadamard) product is novel.

2) Experiments show that such ChebyNet of using high order (1,...,9) Chebyshev polynomials may often improve the precision over the plain networks and be often better than using ordinary polynomials (PolyNet).

**Weaknesses:**

1) The reproducible codes are not provided with the paper anonymously. Since it is mainly an experimental paper for ICLR, reproducible research is necessary to evaluate the experimental results.

2) The motivation of designing ChebyNet architecture seems not clear enough. Among the various possibility of non-adjacent layer interactions, why do the authors choose elementwise product between the sums of Chebyshev polynomials of inputs and features? It seems to me that the 0-th order Chebyshev polynomial recovers the original networks. But what about high order polynomials? Why not additive form? Why not using weighted sum of polynomials while the weights could be tuned?

3) One motivation of exploiting Chebyshev polynomials roughly lies in the fact that their roots, Chebyshev nodes, actually provide a tight bound in polynomial interpolation of continuous functions that minimizes the Runge oscillation phenomenon. Does this property lead to any particular consideration of constructing ChebyNet architecture? Moreover, why does the recursive formulation provide superior numerical stability?

4) In the performance metrics, margins of improvement over the baseline are sometimes small and we are not sure if the improvements are significant. Since this is the first kind of experiments for the proposed methods, it would be better to include certain error bars to account for randomness in evaluations.

5) Figure 5 shows the differences between Cheby-CNN and Poly-CNN in image classification, highlighting the negative correlations on the low orders (0-2) Cheby-CNN. The authors suggest that "The strong correlation among high-order features suggests that low-order features are already sufficient for representing the underlying information, indicating potential for parameter compression". However, from Table 5, middle to high order polynomials seem with high performance as well. Is there a principle in polynomial order selections? By the way, in the last row of this table, some number like 77.1, 76.8 seems missing the highlighted bold font as they are higher than the baseline.

**Questions:**

See the questions raised above in the weakness.

---

### Note · Authors · 2024-11-20

I have read and agree with the venue's withdrawal policy on behalf of myself and my co-authors.